# Safety, efficacy and airway complications of the flexible laryngeal mask airway in functional endoscopic sinus surgery: A retrospective study of 6661 patients

**Chunhua Xi[☯], Dongjing Shi[☯], Xu Cui, Guyan Wang[ID]***

Department of Anesthesiology, Beijing Tongren Hospital, Capital Medical University, Beijing, China

☯ These authors contributed equally to this work.
* guyanwang2006@163.com

**Data Availability Statement:** All relevant data are within the manuscript and its Supporting Information files.

## Abstract

### Objectives

Although the flexible laryngeal mask airway (FLMA) provides considerable advantages in head and neck procedures, little is known about its safety and efficacy in functional endoscopic sinus surgery (FESS). We conducted a retrospective study to evaluate the success rate of FLMA and relevant airway complications in FESS under general anaesthesia.

### Methods

A retrospective review of consecutive patients who underwent FESS for chronic rhinosinusitis was performed from 2015 to 2019. All patients scheduled for FLMA ventilation were identified. Patient characteristics, length of the surgery, FLMA size, failed FLMA cases requiring endotracheal intubation, immediate adverse airway events and delayed airway injuries were recorded. The primary outcomes included the FLMA success rate, which was defined as primary success after induction and final success after the whole surgical procedure. The secondary outcomes were specific clinical factors associated with FLMA failure and airway complications related to FLMA usage.

### Results

Of the 6661 patients included in our study, primary success was achieved in 6572 (98.7%), and final success was achieved in 6512 (97.8%). Failure occurred in 89 patients (1.3%) during induction, in 14 (0.2%) during surgical preparation and in 46 (0.7%) during the intraoperative procedure. All patients with failed FLMA ventilation were successfully switched to endotracheal intubation. Male sex, advanced age, higher American Society of Anesthesiologists grade (ASA) and higher body mass index (BMI) were independent risk factors associated with failed FLMA. Immediate adverse respiratory events were observed in 0.85% of the patients, and delayed airway injuries associated with use of FLMA were observed in 0.07%.

**Funding:** This study was supported by Beijing Hospitals Authority Clinical Medicine Development of Special Funding Support (grant ZYLX202103). The funders had no role in study design, data collection and analysis, decision to publish, or preparation of the manuscript.

**Competing interests:** The authors have declared that no competing interests exist.

## Conclusion

This retrospective study demonstrates a high success rate for FLMA (97.8% in 6661 patients undergoing FESS). Adverse airway events and injuries associated with FLMA are rare, but clinicians should remain vigilant so that early diagnosis and prompt treatment can be provided.

## Introduction

General anaesthesia is performed in most cases of functional endoscopic sinus surgery (FESS), and appropriate anaesthetic management is essential for a successful outcome [1]. Traditionally, endotracheal intubation is the first choice for airway management in FESS because it provides a definite sealing effect on airway protection. The use of a laryngeal mask airway (LMA) is associated with considerable advantages in ear, nose and throat (ENT) procedures, including more stable haemodynamics, a better surgical field, smoother emergence and fewer airway complications than are observed with endotracheal tubes (ETTs) [2]. It has been recently postulated that LMA provides a better surgical field by decreasing catecholamine release in FESS compared to ETT [3]. The flexible laryngeal mask airway (FLMA) is suitable for FESS, as proper placement of the FLMA provides an adequate seal for the oropharyngeal cavity against airway contamination. Compared with other types of LMA and supraglottic airway devices (SGAs), the long flexible and wire-reinforced tube is easily placed under thick layers, stays out of the surgical field, and is resistant to surgical compression, and it does not interfere with surgical manipulation. Additionally, greater movement of the tube during surgery has little effect on the lodgement of the cuff or its ability to seal the airway [4].

However, there is great concern about whether FLMA safely protects the airway during FESS. Manipulation of the well vascularized capillary beds located in the sinonasal mucosa leads to continuous bleeding, and this blood can flow into the glottis and contaminate the tracheobronchial tree during the general anaesthesia and recovery period. Accordingly, many anaesthesiologists and otolaryngologists are reluctant to use FLMA in FESS. However, several prospective trials have indicated that FLMA effectively protects the airway from blood contamination during FESS, showing superiority in protecting the glottis and subglottic airway [5,6]. Other concerns are related to the failure rate of FLMA ventilation, which can occur because, compared with other types of LMA, the elasticity of its tube makes it more difficult to insert and align accurately. FLMA failure is associated with the oropharyngeal structure of the patient, head and neck rotation, insufficient ventilation, gastric insufflation during positive pressure ventilation (PPV), and insufficient anaesthetic depth. Nekhendzy et al reported that the success rate of FLMA was 92.6% in 685 patients who underwent elective ENT surgery; in these patients, half of the failures occurred during induction and half during operation [7]. When FLMA failure occurs during FESS, it is difficult to readjust the position of the FLMA or change to an ETT, and both require interrupting the operation and putting the airway at risk. However, no large sample studies have provided definitive data about FLMA failure and airway complications that occur during FESS.

We conducted this retrospective study at our ternary hospital to evaluate the success rate of FLMA in FESS and to describe the specific clinical factors associated with FLMA failure and airway complications related to FLMA usage.

## Materials and methods

### Ethics approval

The Ethics Committee of Beijing Tongren Hospital, Capital Medical University provided ethical approval for this study on June 20, 2016 (Chairman, Prof. Ningli Wang; protocol number, TRECKY2016-020). The requirement for written informed consent was waived since this is a retrospective study.

### Study design and subjects

We retrospectively reviewed the perioperative clinical information of all adult patients who underwent FESS in our hospital from April 1, 2015, to December 31, 2019. Only patients diagnosed with chronic rhinosinusitis were included. Patients who received FESS for paranasal sinus neoplasms, skull base tumours, and other non-rhinosinusitis were excluded. Other exclusion criteria were emergency surgery, children or adolescents (age under 18), and patients undergoing other neurosurgical, laryngeal and stomatological procedures requiring tracheal intubation. The study period was chosen based on the availability of information from the electronic anaesthesia information system (AIMS, Beijing Easymonitor Tech, Beijing, China) during this period. All preoperative, intraoperative and postoperative data were documented by anaesthesia residents, attending physicians and certified anaesthesia nurses. The patient's medical history (e.g., allergic rhinitis, asthma, and chronic bronchitis), preoperative interview and anaesthesia plan (e.g., airway evaluation, choice of airway device), intraoperative record (e.g., airway device and size, details of airway device changes, ventilation parameters, adverse airway events, time of oral or endotracheal suction and vital signs), and postoperative follow-up in the post-anaesthesia care room (PACU) or in the ward were all collected. We carried out this study and accessed the databases to obtain the data used in this study from July 1, 2016, to January 3, 2020.

The objective of this study was to evaluate the safety and efficacy of FLMA in FESS. The primary outcomes were the success rate of FLMA ventilation, including primary success after induction and final success supporting the whole surgery. The secondary outcomes were (1) specific factors associated with FLMA failure, including patient characteristics, surgical time and FLMA size, (2) adverse airway events, including gastric regurgitation and aspiration, desaturation, bronchospasm, laryngospasm, and airway injuries related to use of FLMA.

Our study was retrospective in nature. To avoid selection bias, we recruited all consecutive patients undergoing elective FESS for chronic rhinosinusitis. In our clinical practice, general anaesthesia for FESS is standardized without being overly particular. Successful ventilation of FLMA in FESS was defined as no audible air leakage from the mouth during manual ventilation at a tidal volume $\geq$ 6 mL/kg, bilaterally identical pulmonary auscultation without auscultation of gastric air insufflation in the epigastric area before the oropharyngeal leak pressure (OLP) arrived, and OLP was at least 5 cm $H_2O$ greater than peak inspiratory pressure (PIP) during PPV. Gastric regurgitation, aspiration, desaturation, upper airway obstruction with/without FLMA displacement, laryngospasm and bronchospasm were defined as immediate airway events since we could diagnose them immediately. Sore throat and hoarseness were evaluated in the PACU. Any patient with a severe sore throat, dysphagia or hoarseness persisting for more than 12 h received fibre-optic laryngoscopy by an otorhinolaryngologist, and any trauma or nerve injury associated with use of FLMA was defined as delayed airway injury.

Clinical information on age, sex, body mass index (BMI), American Society of Anaesthesiologists (ASA) classification, comorbidities, duration of surgery, airway management and airway complications were obtained from the AIMS. Hypercapnia was not always recorded in

our study because end-tidal carbon dioxide partial pressure ($ETCO_2$) was sometimes monitored by portable monitors, and the data were not recorded automatically in the system. Details of FLMA success that occurred during anaesthesia induction or during surgical manipulation were manually reviewed and defined as primary success and secondary success, respectively.

## Anaesthesia management

No premedication was administered. General anaesthesia was induced with intravenous (I.V.) propofol (1.5–2 mg $kg^{-1}$) or etomidate (0.2 mg $kg^{-1}$), sufentanil (0.3 μg $kg^{-1}$) and cisatracurium (0.2 mg $kg^{-1}$), or rocuronium (0.6 mg $kg^{-1}$) and maintained with total intravenous anaesthesia of propofol and remifentanil or inhaled anaesthesia of sevoflurane. FLMA (Medis, Medical Ltd, Tianjin, China) insertion was performed by an anaesthesia resident under the supervision of an anaesthesia attending physician or by the attending physician himself/herself. Each attending physician has performed 500 FLMA procedures annually for at least 3 years. The size of the FLMA was mainly dependent on the patient's weight and based on the manufacturer's recommendation (size 3 for 35–50 kg, size 4 for 50–70 kg, size 5 for 70–100 kg and size 6 for over 100 kg). The FLMA insertion technique is presented in S1 Fig. After insertion, the cuff was inflated to 60 cm $H_2O$. The OLP was checked by setting the pressure-limiting valve of the circular system to 35 cm $H_2O$ and then manually inflating the bag with a fresh gas flow at 5 L $min^{-1}$. The airway pressure was monitored on the anaesthesia machine until the airway pressure reached a steady state and there was air leakage around the patient's mouth. Meanwhile, auscultation over the epigastrium was performed to avoid gastric air insufflation before OLP was achieved. Then, volume-controlled ventilation (tidal volume, 6–8 mL/kg of ideal body weight) was initiated, and the respiratory rate was controlled to maintain $EtCO_2$ at 35–40 mmHg. To ensure the safety of FLMA, the anaesthesia resident was only allowed two attempts at insertion, and then the attending physician was allowed to the third attempt. If FLMA ventilation failed after three insertion attempts during the induction period, the patient was immediately intubated with an ETT. During surgical manipulation, if the PIP abruptly increased or audible air leakage from the mouth occurred, surgical manipulation was suspended, machine-controlled ventilation was changed to manual ventilation, and a muscle relaxant was administered if necessary. If air leakage persisted, oropharyngeal secretions were removed by a suction apparatus, and the patient was intubated with an ETT. Endotracheal suction was subsequently carried out. Details of the change from FLMA to ETT were recorded. FLMA or ETT was removed after the patient regained consciousness or could maintain adequate ventilation (tidal volume of 6–8 ml/kg, respiratory rate ≥10/min and $EtCO_2 < 50$ mmHg). All patients stayed in the PACU for further recovery for more than 20 min.

## Statistical analysis

Statistical analyses were carried out with IBM SPSS 24 software (SPSS Inc., Chicago, IL, USA). Qualitative data are presented as numbers and percentages. Demographic data were evaluated for normality with the Kolmogorov-Smirnov test. Patients with and without FLMA failure were compared according to demographic and clinical characteristics by Pearson's chi-square test or Fisher's exact test for categorical variables, while continuous variables were compared using *t* tests (normally distributed) or Mann-Whitney *U* tests (non-normally distributed). Univariate logistic regressions were performed to identify potentially significant factors. Variables considered to be significant with a *P*-value less than 0.05 in the univariate analysis were entered into the multiple logistic regression analysis.

For the purposes of multivariate analysis, patients were classified according to BMI into underweight ($<18.5$ kg/m$^2$), normal ($18.5$–$22.9$ kg/m$^2$), overweight ($23$–$27.4$ kg/m$^2$) and obese ($\geq 27.5$ kg/m$^2$) groups according to the WHO recommendations for Chinese people [8]; according to age using a continuous concept in which the ranges divided groups into 10-year epochs; and according to ASA dichotomously into low-grade ASA (ASA I) and high-grade ASA (ASA II and ASA III). Because BMI can be considered representative of weight, we excluded weight in the regression model. The adjusted odds ratio (OR) with the 95% confidence interval (95% CI) was used to describe the effect of each independent variable on the risk of FLMA failure.

Sample size estimation was performed to define the power of the data analysis as previously described [9,10]. Based on an expected FLMA failure rate of 2.3%, a sample size of 4347 would allow the evaluation of at least 10 covariates (five were included in our study).

## Results

A total of 6790 consecutive adult patients who underwent FESS for chronic rhinosinusitis were identified from 2015 to 2019. Among them, 129 were preoperatively scheduled to receive an ETT, including 63 for additional pharynx surgery, 21 for exodontia, 3 for neurosurgery, 3 due to potentially difficult airways, 4 due to severe impairment in lung function requiring possible postoperative ventilation support, 18 requiring a long surgery time ($>4$ h), and 17 with no detailed reasons. All patients scheduled for ETT were excluded from this study. The remaining 6661 patients were scheduled for FLMA, among whom 89 (1.3%) experienced primary failure after induction, with a primary success rate of 98.7%. Sixty (0.9%) patients experienced secondary failure during surgical manipulation: 14 (0.2%) occurred during surgical preparation when the position of the patient's head and neck was changed, and 46 (0.7%) occurred during the intraoperative procedure. Final success was achieved in 6512 (97.8%) patients (Fig 1). ETT was successfully performed for all cases of failed FLMA.

The demographic and clinical data of the patients, including age, sex, BMI, ASA classification, and comorbidities, are shown in Table 1. In the univariate analysis, we found that the following six variables contributed to a higher incidence of failed FLMA: male sex, age, weight, BMI, ASA grade and LMA size. Apart from weight, five other variables were included in the logistic regression analysis, and four independent predictors of FLMA failure were identified: male sex (OR 3.02, 95% CI 1.83–4.86), age (OR 2.23, 95% CI 1.45–3.43), ASA grade (OR 1.58, 95% CI 1.38–1.81) and BMI (OR 1.63; 95% CI, 1.24–2.13) (Table 2).

Of the 6661 patients, 56 (0.85%) experienced immediate respiratory airway events (Table 3). Desaturation occurred before, during and after the surgery in 10 (0.15%), 9 (0.14%) and 37 (0.56%) patients, respectively. Laryngospasm occurred in 24 (0.36%) patients during recovery, and among these patients, two suffered negative pressure pulmonary oedema (NPPE). Both patients required immediate ETT and intensive care management. There were 5 (0.08%) cases of bronchospasm, of which two occurred during induction and mask ventilation and three occurred during recovery after the FLMA was removed. During the surgical procedure, no episodes of regurgitation or pulmonary aspiration were observed. During postoperative evaluation, 636 (9.55%) patients experienced minor sore throat; however, recurrent nerve injury, arytenoid dislocation and pharyngopalatine arch damage were also observed in 1, 2 and 2 patients, respectively, and the cumulative rate of delayed airway injury was 0.08%.

## Discussion

In this retrospective study, we demonstrated that of 6661 patients undergoing FESS for chronic rhinosinusitis, the incidence of FLMA final success was 97.8%, and the intraoperative failure

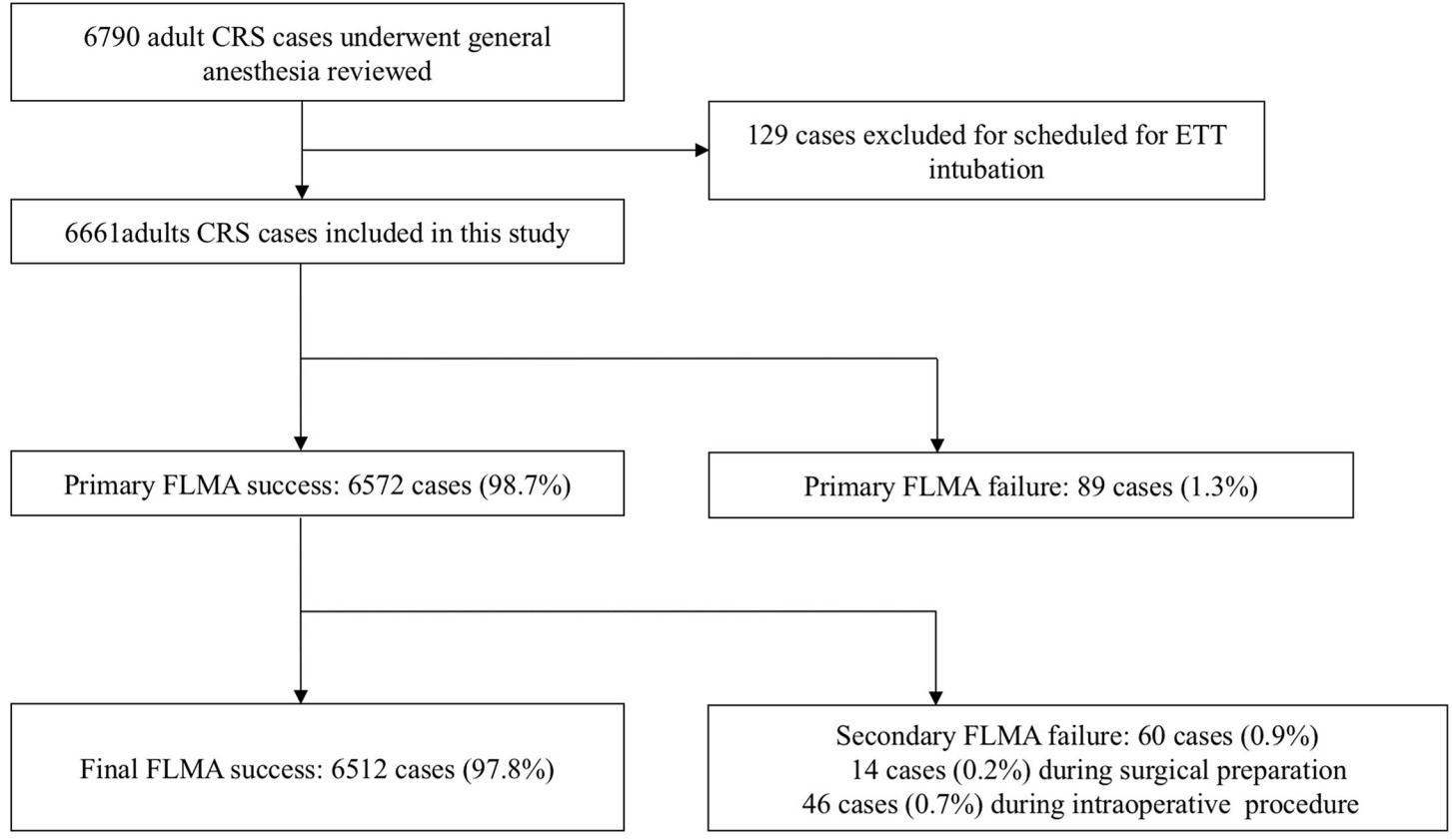

**Fig 1. Study, methodology and main results in FLMA airway management.** CRS = chronic rhinosinusitis; FLMA = flexible laryngeal mask airway.

rate was 0.7%. Four independent factors were found to predict FLMA failure: male sex, advanced age, higher ASA grade and increased BMI. Immediate adverse respiratory events were observed in 0.85% of cases, and the incidence of delayed airway injuries associated with use of FLMA was 0.07%.

## Safety and efficacy of FLMA

One of the great concerns regarding the application of FLMA in FESS is whether it can protect the airway from surgical bleeding and flushing fluid during general anaesthesia with PPV. OLP is a critical parameter to evaluate the sealing function of LMA. A higher OLP means better sealing and little possibility of air leakage. Many factors contribute to OLP, including the LMA position, type and size of the LMA, cuff pressure, position of the head and neck, surgical manipulations and usage of muscle relaxants [11,12]. Although an OLP test is recommended before surgery, there is so far no consensus regarding the specific OLP value that is safe for the airway. Another problem is the question of how inadvertent gastric insufflation can be avoided because the high airway pressure delivered by the LMA might be transmitted to the oesophagus and stomach and could increase the possibility of regurgitation and aspiration. Usually, gastric insufflation is detected by a stethoscope placed at the epigastric area [13]. Gastric insufflation occurs more when the PIP is increased during PPV [14]. Therefore, a lower PIP is helpful to reduce the incidence of gastric insufflation. In our clinical practice, we ensured that all FESS cases with FLMA usage were compliant in maintaining a difference between OLP and PIP of more than 5 cm $H_2O$, and no air leakage was detected around the epigastric area before

**Table 1. Patient demographics according to successful versus failed LMA.**

|  | Total | Success | Failure | p-value* |
|---|---|---|---|---|
| **Patients (n [%])** | 6661 (100) | 6512 (97.8) | 149 (2.2) | - |
| **Male sex (n [%])** | 3984 (59.8) | 3849 (59.1) | 116 (77.8) | <0.001 |
| **Age (years)** | 47 (35–57) | 44 (32–55) | 58 (48–62) | <0.001 |
| **Height (cm)** | 170 (162–175) | 170 (162–175) | 170 (164–174) | 0.711 |
| **Weight (kg)** | 70 (60–80) | 70 (60–80) | 75 (62–85) | <0.001 |
| **BMI (kg/m$^2$)** | 24.4 (22.1–26.8) | 24.4 (22.1–26.7) | 26.7 (22.7–29.4) | <0.001 |
| **LMA size** |  |  |  | 0.008 |
| **3** | 49 (0.7) | 49 (0.7) | 0 (0) |  |
| **4** | 3166 (47.5) | 3112 (47.8) | 54 (36.2) |  |
| **5** | 3418 (51.3) | 3325 (51.1) | 93 (62.4) |  |
| **6** | 28 (0.4) | 26 (0.4) | 2 (1.3) |  |
| **Comorbidity** |  |  |  |  |
| **Asthma (n [%])** | 566 (8.5) | 554 (8.5) | 12 (8.0) | 0.962 |
| **Hypertension (n [%])** | 969 (14.5) | 939 (14.4) | 30 (20.1) | 0.066 |
| **CAD (n [%])** | 187 (2.8) | 179 (2.7) | 8 (5.4) | 0.096 |
| **ASA grade** |  |  |  | <0.001 |
| **I** | 3003 (45.1) | 2975 (45.6) | 28 (18.7) |  |
| **II** | 3417 (51.3) | 3312 (50.9) | 105 (70.5) |  |
| **III** | 241 (3.6) | 225 (3.5) | 16 (10.7) |  |
| **Surgical time (min)** | 67 (44–101) | 67 (45–101) | 72 (48–100) | 0.145 |

Data are presented as n (%) or median (25th and 75th percentile). All 6661 patients were included in the regression model.

*P values were calculated by Chi-square or Fisher's exact test for categorical variables and the Mann-Whitney U tests for nonnormally distributed continuous variables.

FLMA = flexible laryngeal mask airway; FESS = functional endoscopic sinus surgery; ASA = American Society of Anaesthesiologists Physical Status; BMI = body mass index, CAD = coronary artery disease.

OLP was achieved. PIP was carefully monitored during surgical manipulation, and any abruptly increased PIP or air leakage was handled immediately. Among the 6661 patients in our study, no severe respiratory event was associated with FLMA failure or an inability to protect the airway. This result supports our strategy of controlling the gap between OLP and PIP, which plays an important role in ensuring the safety of FLMA during PPV. Further studies aimed at exploring the advantages of our strategy could focus on respiratory dynamics, air leakage and gastric insufflation pressure. However, gastric insufflation is not continuously

**Table 2. Independent risk factors for FLMA failure during FESS.**

| Risk Factor | P Value* | Adjusted Odds Ratio | 95% Confidence Interval | |
|---|---|---|---|---|
|  |  |  | Lower | Upper |
| **Male sex*** | <0.001 | 3.02 | 1.87 | 4.86 |
| **Age *** | <0.001 | 2.23 | 1.45 | 3.43 |
| **ASA*** | <0.001 | 1.58 | 1.38 | 1.81 |
| **BMI (kg/m$^2$)*** | <0.001 | 1.63 | 1.24 | 2.13 |
| **LMA size** | 0.090 | 0.66 | 0.41 | 1.07 |

*Any variable with a P value of less than 0.05 was considered an independent predictor of failed FLMA.

FLMA = flexible laryngeal mask airway; FESS = functional endoscopic sinus surgery; ASA = American Society of Anaesthesiologists Physical Status; BMI = body mass index.

**Table 3. Incidence of respiratory airway events during FESS.**

| Respiratory airway events | n | (%) |
|---|---|---|
| **Perioperative desaturation** | 10 | 0.15% |
| Bronchospasm | 2 | 0.03% |
| Upper airway obstruction | 8 | 0.12% |
| **Intraoperative desaturation** | 9 | 0.14% |
| FLMA displacement | 9 | 0.14% |
| **Postoperative desaturation** | 37 | 0.56% |
| Laryngospasm | 24 | 0.36% |
| Bronchospasm | 3 | 0.05% |
| Upper airway obstruction | 10 | 0.15% |
| **Total** | **56** | **0.85%** |

Desaturation was defined as SPO2 $\leq$ 90% on two consecutive readings with an interval of 60 s.

monitored during the operation in our clinical practice. Thus, asymptomatic gastric insufflation might be underestimated in our study. Nonetheless, we consider that FLMA with a gastric drainage aperture is a better choice for FESS.

Since the FLMA is more flexible, its insertion may be more difficult than that of a classic LMA [4]. The total incidence of FLMA failure of 2.2% in our study is slightly higher than the failure rates of 0.86–1.1% for LMA reported in other large observational non-ENT studies [9,10] and significantly lower than the 7.4% failure rate previously reported for FLMA in ENT surgery [7]. Our relatively low failure rate might be attributed to our clinical practice of having an attending supervise. We did not count the number of attempts at FLMA insertion because the number depends on the experience of the anaesthetist. Our success rate is attributed to the level of experience of the attending physician, which precludes experience as a confounder in this study.

The physiological characteristics of the patient, such as male sex, advanced age, and high BMI, were associated with FLMA failure in our study, consistent with previous reports on non-ENT surgery [9,15]. Patients with a high BMI are likely to have larger pharyngeal fat pad areas [16], which leads to a decrease in the pharyngeal area. A narrow pharyngeal cavity increases the number of FLMA attempts, the failure rate and the incidence of airway injury. Ageing increases pharyngeal muscular atrophy and pharyngeal volume and flattens the upper airway [17]; these anatomical changes in the pharyngeal cavity might influence the success of FLMA. ASA grade was also identified as an independent predictor of FLMA failure, a result that was likely attributable to the associations between higher ASA grade and both advanced age and overweight/obesity in our study. It is interesting that male sex was the risk factor most significantly associated with FLMA failure. This result could be explained by the fact that certain craniofacial features specific to males affect the oropharyngeal cavity and make them prone to upper airway obstruction during sleep and general anaesthesia [18].

With regard to secondary failure that occurred during the surgical procedure, we consider that both surgical manipulation of the head and neck and insufficient muscle relaxation could lead to an abrupt increase in PIP, FLMA displacement and FLMA ventilation failure. Before the surgical procedure, surgical preparations include covering over the patient's head with a sterile treatment sheet, which requires flexing the patient's head for a while; at the beginning of the surgery, the surgeon usually rotates the patient's head 10 to 15 degrees towards him to facilitate surgical manipulation. Previous studies report that the flexed neck position may elicit airway obstruction and impair ventilation, whereas rotation of the head and neck has little effect

on supraglottic airway device ventilation [19,20]. In our study, fourteen cases (0.2%) failed after changing the head and neck position; however, forty-six cases (0.7%) failed intraoperatively, when the position of the patient's head was rarely changed. Therefore, we suggest that changing the head and neck position is not the only reason for secondary failure. Consistent with other reports, we found that almost half of the FLMA procedures performed without an optimal fibreoptic bronchoscopy (FOB) position provided satisfactory OLP and ventilation [13,21]. We therefore consider that if the FLMA is sub-optimally placed in the oesophageal inlet, displacement is apt to occur when the patient's swallowing reflex recovers or their head and neck position changes, resulting in FLMA dislodgement and ventilation failure. Further studies should focus on whether sub-optimal positioning of the FLMA is related to secondary failure during surgery.

### Airway complications of FLMA during FESS

The incidence of adverse respiratory events was 0.85% in our study, consistent with a previous report on non-ENT surgery [9]. Nineteen (0.28%) patients experienced desaturation during induction and the operation, which could be explained by FLMA failure and a change in airway device; however, other complications, such as bronchospasm during mask ventilation, laryngospasm and bronchospasm after FLMA removal, might not be directly related to the use of FLMA.

Compared with other surgical procedures, otolaryngical procedures are associated with a high incidence of perioperative respiratory adverse events in both children and adults [22,23]. For example, intraoperative bronchospasm occurs more frequently during sinus surgery than during nonsinus surgery, with an incidence of 0.7 to 2.6% [24]. The possible explanations include direct surgical stimulation of the airway and a higher incidence of airway hypersensitivity. It has been recognized that in sinonasal diseases, upper airway remodelling overlaps with lower airway remodelling [25,26]; therefore, patients with chronic rhinosinusitis have a high incidence of coexisting allergic rhinitis and asthma and are thus at risk of perioperative bronchospasm and laryngospasm. For these patients with allergic rhinitis and asthma, a thorough preoperative assessment including patient history, physical examinations and lab studies should be performed to ensure that the airway disease is well controlled. Since tracheal intubation is a well-known trigger of bronchospasm [27], FLMA is our first choice to control the airway for chronic rhinosinusitis unless there are other contraindications. The FLMA should be removed when these patients are fully conscious to ensure adequate gastroesophageal reflux and avoid aspiration. Despite these precautions, five (0.08%) cases of life-threatening severe bronchospasm attacks occurred during induction and recovery. We suggest that more effort is needed to identify the potential risks of bronchospasm in patients who undergo otolaryngical procedures, and guidelines for early identification and prompt management would be valuable to prevent further deterioration.

In our study, postoperative laryngospasm was the most common airway complication. Systematic reviews have shown that there is no clear difference in the incidence of postoperative laryngospasm between LMA and tracheal tubes in adults [2,28]. The direct reasons for postoperative laryngospasm were not well documented in our study. Based on our experience, insufficient suction of blood and inappropriate timing of FLMA removal can lead to laryngospasm and result in severe complications. In this study, laryngospasm occurred in two healthy muscular male patients when they unconsciously bit the FLMA during recovery. Refractory desaturation and NPPE occurred. The incidence of NPPE is approximately 0.1% in adults undergoing general anaesthesia [29]. Although NPPE is more closely associated with tracheal extubation, an increase in LMA-related NPPE has been recently reported due to the higher

proportion of procedures involving LMA applications [30,31]. In our study, the development of NPPE might be explained by forceful inspiration and acute airway obstruction due to laryngospasm. We encourage our colleagues to be vigilant in monitoring for refractory desaturation during recovery and aware of the occurrence of and treatment for NPPE while using LMA. Additionally, we suggest that the FLMA should be removed when the patient is fully conscious after FESS.

Considering other complications, the 9.55% incidence of sore throat observed in our study is within the range of previous reports (5.8 to 34%) [32]. It should be noted that the incidence of severe nerve and tissue damage related to FLMA usage, including arytenoid dislocation, pharyngopalatine arch damage and hoarseness caused by recurrent laryngeal nerve injury, was 0.07% in our study. The patients' medical records showed that all these injuries were associated with repeated FLMA insertion. All of the affected patients underwent unplanned therapy, and no sequelae occurred. We suggest that more attention should be paid to improving the success of first insertion and avoiding excessive attempts.

## Limitations

Our study has some limitations that should be noted. First, this is a retrospective study, which includes the lack of randomization and exclusion of some confounding factors. However, FLMA management is standardized in our hospital, and we recruited all consecutive patients undergoing FESS for chronic rhinosinusitis within a certain period. The results of this study exclude subjective factors as much as possible. Second, we did not count the number of attempts for each case because of incomplete data, and we do not routinely use FOB to assess the FLMA position or carry out neuromuscular monitoring for each case; therefore, we cannot provide more information about the cases of failed FLMA. Third, gastric insufflation was excluded after FLMA insertion, but it was not continuously monitored during the operation; thus, asymptomatic gastric insufflation might be underestimated in our study. Fourth, patients with chronic rhinosinusitis are likely to suffer from obstructive sleep apnoea (OSA), possibly due to increased upper airway inflammation and increased upper airway resistance [33,34]. Although many patients reported symptoms of OSA in their medical records, not all of these cases were confirmed by overnight polysomnography. Thus, OSA was not included as an independent variable in our univariate analysis. Finally, our study was carried out in a single tertiary health centre, but otorhinolaryngology is a key discipline in our hospital. Both the surgeon and the anaesthesiologist are familiar with FLMA and each other's surgical approaches. Differences in anaesthetic and surgical techniques should be considered when interpreting our results.

Despite these limitations, the results of our study shed light on a common but little-studied issue encountered by both anaesthesiologists and rhinologists. By demonstrating the FLMA failure rate, high-risk factors for FLMA failure, and the incidence of airway complications in a large patient sample, this study provides detailed evidence that can be used by clinicians when deciding whether to use FLMA in FESS.

## Conclusions

In this study, we report a FLMA success rate of 97.7% in 6661 adult patients undergoing elective FESS for chronic rhinosinusitis. Intraoperative FLMA failure was uncommon (0.7%). Male sex, advanced age, higher ASA grade and higher BMI were independent risk factors for failed FLMA ventilation. FLMA-related adverse respiratory events and airway injures are rare, but clinicians should remain vigilant so that early diagnosis and prompt treatment can be provided.

## Supporting information

**S1 Fig. The technique of FLMA insertion.** (A) Opening the oral cavity; (B) Placing the laryngeal mask into the oral cavity; (C) Lacing the laryngeal mask into the pharyngeal cavity; (D) FOB view.
(TIF)

## Acknowledgments

Assistance with the article: We gratefully acknowledge Shaofei Su, PhD (Department of Epidemiology and Health Statistics, Beijing Obstetrics and Gynaecology Hospital, Capital Medical University) for statistical consultation and the editors at American Journal Experts for their assistance in improving the English language herein.

## Author Contributions

**Data curation:** Chunhua Xi, Dongjing Shi.

**Formal analysis:** Chunhua Xi, Xu Cui.

**Investigation:** Chunhua Xi, Dongjing Shi.

**Methodology:** Chunhua Xi, Dongjing Shi, Xu Cui, Guyan Wang.

**Resources:** Chunhua Xi.

**Supervision:** Xu Cui, Guyan Wang.

**Writing – original draft:** Chunhua Xi, Dongjing Shi.

**Writing – review & editing:** Xu Cui, Guyan Wang.

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
