## [Decision Letter · Decision Letter 0]

21 Sep 2020

PONE-D-20-20651

Safety, efficacy and airway complications of the flexible laryngeal mask airway in functional endoscopic sinus surgery: a retrospective study of 6661 patients

PLOS ONE

Dear Dr. Wang,

Thank you for submitting your manuscript to PLOS ONE. After careful consideration, we feel that it has merit but does not fully meet PLOS ONE’s publication criteria as it currently stands. Therefore, we invite you to submit a revised version of the manuscript that addresses the points raised during the review process.

ACADEMIC EDITOR

Thank you for your submission that retrospectively analyses the use of flexible laryngeal mask airways in a large cohort of patients undergoing FESS. To my knowledge this is the largest study of its kind and in this lies the real strength of the data obtained. The major concern of this paper are the biases and limitations associated with it being retrospective in nature. Although this is acknowledged in the limitation section of the paper I would also recommend elaborating on the specific bias ( e.g recall, and selection bias etc) that is found in retrospective observational reports. Unless this study recruited all consecutive patients undergoing FESS, is it not possible that FESS patients deemed to be at high risk of failing LMA were excluded from the study which would therefore influence the true failure rate. Please address the above and the questions and suggestions raised by both reviewers

We look forward to receiving your revised manuscript.

Kind regards,

Alkis James Psaltis, PhD, MBBS(HONS), FRACS

Academic Editor

PLOS ONE

2.PLOS requires an ORCID iD for the corresponding author in Editorial Manager on papers submitted after December 6th, 2016. Please ensure that you have an ORCID iD and that it is validated in Editorial Manager. To do this, go to ‘Update my Information’ (in the upper left-hand corner of the main menu), and click on the Fetch/Validate link next to the ORCID field. This will take you to the ORCID site and allow you to create a new iD or authenticate a pre-existing iD in Editorial Manager. Please see the following video for instructions on linking an ORCID iD to your Editorial Manager account: https://www.youtube.com/watch?v=_xcclfuvtxQ

4. We note that Figure [S1] includes an image of a patient / participant in the study. 

<h1>** **</h1>

Reviewers' comments:

Reviewer's Responses to Questions

**Comments to the Author**

1. Is the manuscript technically sound, and do the data support the conclusions?

Reviewer #1: Yes

Reviewer #2: Partly

2. Has the statistical analysis been performed appropriately and rigorously? 

Reviewer #1: I Don't Know

Reviewer #2: I Don't Know

3. Have the authors made all data underlying the findings in their manuscript fully available?

Reviewer #1: Yes

Reviewer #2: Yes

4. Is the manuscript presented in an intelligible fashion and written in standard English?

Reviewer #1: Yes

Reviewer #2: No

5. Review Comments to the Author

Reviewer #1: The authors present a retrospective analysis comprising 6661 patients who underwent FESS for chronic rhinosinusitis with a FLMA. The study aims to evaluate the safety and efficiency of FLMA in FESS. Furthermore, risk factors for FLMA failure and airway complications are evaluated. The authors conclude that the FLMA can be safely and effectively used in patients undergoing FESS. The manuscript is written very well, includes a high number of patients and addresses an interesting question.

Other studies have shown outcome of the FMLA use in ENT surgery, so the impact of the study regarding the raised question might be limited. However, these are still significant findings as it is the first report of this kind specifically for this subgroup of ENT patients. Particularly endoscopic sinus surgery can lead to significant bleeding the question of whether this kind of airway management in these patients is safe is of interest. Therefore, it is a valuable scientific contribution and should be published. However, there are some severe and minor concerns which require a major revision as follows.

- was the insertion and application of the FMLA also performed by anesthetists in training? Or might the high success rate compared to other studies (7,2% in NeKHeNDZY et. al) be attributed to the high level of experience of senior anesthetists. This might be a confounder of the study.

- Regarding risk factors it is surprising that the BMI is associated with adverse outcome.The authors should discuss the fact that others did not find an association between primary success/failure and patients’ age and BMI.

- Were the number of insertion attempts documented and evaluated as a predictor for failed FLMA application? If not the authors should include this in the analysis.

- How was patient selection performed? Were all patients, even those with highest ASA scores and comorbidities part of the analysis or primarily attributed for EET?

- One of the major concerns of the use of FMAL is gastric insufflation. Malpositioning might occur despite seemingly uneventful FMLA placement. What was the percentage rate of gastric insufflation in the study cohort?

- From the FESS surgeons perspective the rotation of the patient’s head is crucial to achieve good surgical outcomes. Was rotation of the patient’s head the most the direct reason for secondary failure?

- The authors should elaborate and extend the section on advantages of FMLA to underline the key adavantages of the FMLA over EET. Furthermore, a clarification of the main differences between the FMLA and the conventional laryngeal mask should be included as the non-anesthetist readership might not be completely aware why the conventional mask is not applicable in FESS.

Reviewer #2: the manuscript claims to report safety, efficacy and airway complications of the flexible laryngeal mask airway in functional endoscopic sinus surgery as retrospective study. The strength of the study is author’s efforts in conducting a large retrospective study. We commend them to perform this study in a single centre. The weakness is the flaw in the methodology and flow of language and content. It needs a major revision in various areas of the article.

Summary: the manuscript claims to report safety, efficacy and airway complications of the flexible laryngeal mask airway in functional endoscopic sinus surgery as retrospective study. The strength of the study is author’s efforts in conducting a large retrospective study. We commend them to perform this study in a single centre. The weakness is the flaw in the methodology and flow of language and content. It needs a major revision in various areas. The main points are highlighted below.

Abstract:

In the objective section the primary aim is not mentioned but states safety and efficiency.

What is the main objective? The authors should clarify it.

In methods though Primary outcome was the success rate, the secondary aim does not specify clinical factors which are related to the failure. Authors needs to be clarify it to avoid confusion

Conclusion: Instead of reporting the results of primary outcome, authors conclude on FLMA safety and efficacy. In the second sentence the author’s mention that the intra operative failure is uncommon. But failed to the failure rate. I don’t know what the latter half of the sentence means: “detected quickly and handled immediately”. The data or text in the results doesn’t support it. Specific comments below

Methods: What is EEG?

Methodology is inadequate. Inclusion criteria is only the time period of data collected but doesn’t give other details. Inclusion or exclusion criteria doesn’t mention on patients who had FESS with ETT.

In results in line 145 the some details were given. Example: 129 (1.9%) were scheduled for ETT .

Then line 88: what I can understand is the departmental standardised GA options. It is not clear whether it’s their departmental protocol. If it is, whether it was applied in all cases by all the anaesthetists. Doesn’t mention on non-availability of data in exclusion criteria.

Line 104: can’t understand what authors want to say. What is lead around the patient neck?

Results: Line 144,149,-153: 6661 patients were initiated and final success of FLMA was 6512.

The number do not add up to the figure of 6512. Based on the failures (89+60+14+46=209) it should be 6452. Please check.

Table 1 only one p value (0.008) was given for all LMAs. Is it different for all sizes?

Similarly for ASA grade, does one p value <0.001 applies to all grades?

Table 3 the total respiratory adverse events given as (n) 56 nerve. Adverse events under this subheading are they different to above total? If it is 56 then events mentioned do not add up. The subheading on tissue damage adverse events are immediate and delayed complications, suggest change of wording.

Discussion: under safety and efficacy: did all cases were compliant in maintaining the OLP and PIP difference of 5H20. Line 226 mentions inadequate relaxation, whether neuromuscular monitoring was carried out?

Line 227- 233: Author’s mentions sometimes reducing tidal volume and increasing respiratory frequency were required. Is it author’s opinion on its management or speculation what occurred?

The line 246: previous study is on prevalence, risk factors, and management of asthma in China: a national cross-sectional study. I don’t understand the relevance here as this is FESS study assessing the safety and efficacy of the airway.

Awake FLMA: what’s the relevance in your retrospective study? Has the authors conducted any such procedures?

Line 260 and 261 contradicts. If not documented in the medical chart, then is it author’s opinion?

In results there were two patients had laryngospasm had NPPE. Describing their management in discussion may not be necessary?

Line 281: says no life threatening damage, but authors mentioned 5 such cases. It contradicts the previous statement.

Limitations paragraph is mixed with limits and many improvement suggestions for the future. It’s confusing to the readers.

Line 301: ranks top: superlative may be avoided

Conclusion: please refer to the comment in the abstract.

References: adequate

6. PLOS authors have the option to publish the peer review history of their article (what does this mean?). If published, this will include your full peer review and any attached files.

Reviewer #1: No

Reviewer #2: No

---

## [Author Response · Author response to Decision Letter 0]

6 Nov 2020

Response to Reviewers

Dear Editor, 

I greatly appreciate you and the reviewers for all the comments, critiques and efforts to help optimize our manuscript. We have made all of the changes you recommended and ensured that all of the documents now correspond to the correct format. As previously communicated, we are grateful that the reviewers raised interesting issues, which we believe will help better communicate our message, and we are happy to respond to their comments. 

Our point-by-point responses are listed below.

Handling Editor’s Comments:

1. The major concern of this paper are the biases and limitations associated with it being retrospective in nature. Although this is acknowledged in the limitation section of the paper I would also recommend elaborating on the specific bias ( e.g recall, and selection bias etc) that is found in retrospective observational reports. Unless this study recruited all consecutive patients undergoing FESS, is it not possible that FESS patients deemed to be at high risk of failing LMA were excluded from the study which would therefore influence the true failure rate. Please address the above and the questions and suggestions raised by both reviewers

Yes, this study recruited all consecutive patients undergoing FESS for chronic rhinosinusitis from April 1, 2015 to December 31, 2019. We mentioned this point in abstract (line 25), method (line 91), and result (line 184). From line 113 to line 131 in method section, we described the data collection and definition of our outcome in detail to avoid selection bias.

Reviewers' Comments:

Reviewer #1: 

 1.was the insertion and application of the FMLA also performed by anesthetists in training? Or might the high success rate compared to other studies (7,2% in NeKHeNDZY et. al) be attributed to the high level of experience of senior anesthetists. This might be a confounder of the study.

We didn’t count the attempt times of FLMA insertion, and we only mention successful ventilation in our study. Attempt times might reflect the experience of the anesthetists. In our clinical practice, general anaesthesia for FESS is standardized without being very particular. In our study, FLMA (Medis, Medical Ltd, Tianjin, China) insertion was performed by an anesthesia resident under supervision of an anesthesia attending physician, or by an attending himself/herself. Each attending physician has performed FLMA 500 cases annually at least for 3 years (line 137 to 140). To ensure the safety of FLMA usage, the anaesthesia resident was only allowed two attempts at insertion, and then the attending physician was allowed to try for a third time (line 149 to 151). If the third time is not successful, we change to endotracheal intubation(ETT). Our final success rate is attributed to the attending physician. We consider that the difference between our results and others (7.2% in NeKHeNDZY et. al) is our clinical practice (attending physician supervision, strict control of the difference between OLP and PIP), and we only recruited FESS without facial plastic surgery (in NeKHeNDZY et. al)). In our experience, patients who needs facial plastic surgery usually combined with maxillofacial dysplasia, a potential risk of difficult airway and LMA failure, or need more surgical manipulation of head and neck. ( NeKHeNDZY et. Al didn’t describe this detail).

2. - Regarding risk factors it is surprising that the BMI is associated with adverse outcome. The authors should discuss the fact that others did not find an association between primary success/failure and patients’ age and BMI.

Yes, it is interesting that few article has discussed the relation between FLMA failure and BMI/age. In our hospital, our otolaryngologis use upper airway computed tomography to evaluate the effect of aging and high BMI on upper airway morphology changes[1]. We’ve discussed the result of our study together, and they’ve pointed out the effect of upper airway morphology change and FLMA lodgement. We’ve added this discussion from line 272 to 276. 

 3. Were the number of insertion attempts documented and evaluated as a predictor for failed FLMA application? If not the authors should include this in the analysis.

We didn’t count the attempt times of FLMA insertion, because attempt times depends on the experience of the anaesthetists. Our final success rate is attributed to the level of experience of the attending physician, which excludes the experience as a confounder in this study. We discuss this point from line 266 to line 269.

4. - How was patient selection performed? Were all patients, even those with highest ASA scores and comorbidities part of the analysis or primarily attributed for EET?

The description of method was not very clear in our primary manuscript, and we have revised then.

Patients’ recruitment is described in method section from line 91 to line 96 and result section from line 184 to line 189. Patients with severe impairment in lung function requiring possible postoperative ventilation support was primarily attributed for EET, and this was documented in the preoperative evaluation and we can manually review it and exclude it. FESS for chronic rhinosinusitis is an elective surgery. Patient with highest ASA scores and comorbidities, or with a poor general condition is not appropriate for this surgery. Patients with ASA III, who have stable coronary disease or chronic heart function insufficiency are always scheduled for FLMA, unless they have severe respiratory impairment. Urgent surgery, such as hemostasis for epistaxis, is not included in our study.

5- One of the major concerns of the use of FMAL is gastric insufflation. Malpositioning might occur despite seemingly uneventful FMLA placement. What was the percentage rate of gastric insufflation in the study cohort?

We didn’t find any gastric regurgitation or aspiration related to gastric insufflation. In our practice, we ensure that there was no gastric insufflation before oropharyngeal leak pressure (OLP) was arrived by auscultation over the epigastrium, and make sure that the oropharyngeal leak pressure (OLP) is at least 5 cm H2O greater than peak inspiratory pressure (PIP) during Positive pressure ventilation to avoid air leakage or gastric insufflation.(method section, from line 115 to line 119) (For a patient with normal BMI, the PIP is usually around 12 to 16 cm H2O, and the OLP is more than 24 cm H2O, so this strategy is realistic). However, gastric insufflation was not continuously monitored during the operation in our clinical practice. Asymptomatic gastric insufflation might be underestimated in our study, and we consider that the FLMA with a gastric drainage aperture is a better choice for FESS. We discuss this point from line 259 to line 262.

6- From the FESS surgeons perspective the rotation of the patient’s head is crucial to achieve good surgical outcomes. Was rotation of the patient’s head the most the direct reason for secondary failure?

Yes, the rotation of the patient’s head is crucial to provide better surgical view for FESS surgeons. Before the surgical procedure, the surgical preparation of covering over the patient's head with a sterile treatment sheet needs to lift the patients head. The flexed neck position decreases the space of oropharyngeal cavity and might impact the LMA ventilation[2, 3]. After disinfection and surgical drape, the surgeon usually rotates the patient's head 10 to 15 degrees towards him to facilitate the surgical manipulation. 14(0.2%) secondary FLMA failure happened in this period in our study. We performed ETT immediately without interrupt the surgical manipulation during this period. In our clinical practice in general anesthesia for ear surgery, which should rotate the patient's head 70 degrees to the opposite side, together with others report, we’ve found that this rotation does not impact FLMA ventilation if the FLMA is in the optimal position. So we consider that if the FLMA has a suboptimal lodgement in the oesophageal inlet, displacement is apt to occur when the swallowing reflex recovers or the head and neck position changes, resulting in FLMA dislodgement and ventilation failure. Further studies should focus on whether suboptimal positioning of the FLMA is related to secondary failure during surgery (line 294 to line 300).

7. - The authors should elaborate and extend the section on advantages of FMLA to underline the key adavantages of the FMLA over EET. Furthermore, a clarification of the main differences between the FMLA and the conventional laryngeal mask should be included as the non-anesthetist readership might not be completely aware why the conventional mask is not applicable in FESS.

We added more detailed information to describe FLMA and its advantage in FESS compared to ETT and conventional laryngeal mask in the first paragraph (line 54 to line 62)

Reviewer #2:

1. 

1.1 Abstract:In the objective section the primary aim is not mentioned but states safety and efficiency.What is the main objective? The authors should clarify it.

In abstract, we clarify our main objective, that is we conducted a retrospective study to evaluate the success rate of FLMA and relevant airway complications in FESS under general anaesthesia (line 23 to line 24).

1.2. in methods though Primary outcome was the success rate, the secondary aim does not specify clinical factors which are related to the failure. Authors needs to be clarify it to avoid confusion

We clarify the clinical factors from line 27 to 29

1.3 Conclusion: Instead of reporting the results of primary outcome, authors conclude on FLMA safety and efficacy. In the second sentence the author’s mention that the intra operative failure is uncommon. But failed to the failure rate. I don’t know what the latter half of the sentence means: “detected quickly and handled immediately”. The data or text in the results doesn’t support it.

We’ve changed our conclusion. We’ve confirmed our successful rate in the first sentence. This is our first time to report a retrospective study, and we are easy to mix our results and our subjective view. We’ve made a better revision of the manuscript.

2. Methods: What is EEG? 

This sentence has been cancelled.

3. Methodology is inadequate. Inclusion criteria is only the time period of data collected but doesn’t give other details. Inclusion or exclusion criteria doesn’t mention on patients who had FESS with ETT. 

From line 92 to line 132, we’ve almost re-organized the method section. We’ve made the inclusion and exclusion criteria clearer in detail, and we clarify the primary outcome and secondary outcome. We’ve described the method of data reservation and collection. 

4. In results in line 145 the some details were given. Example: 129 (1.9%) were scheduled for ETT. Then line 88: what I can understand is the departmental standardised GA options. It is not clear whether it’s their departmental protocol. If it is, whether it was applied in all cases by all the anaesthetists. Doesn’t mention on non-availability of data in exclusion criteria.

In method section, from line 116 to line 118 in method section, We’ve described that general anaesthesia for FESS is standardized without being very particular, and we’ve described our anesthetic management in detain from line 136 to line 163. Data in exclusion criteria was list in in method section from line 93 to line 98 and result section from line 187 to line 193. 

5. Line 104: can’t understand what authors want to say. What is lead around the patient neck?

It has been revised to “air leakage around the patient’s mouth.”

6. Results: Line 144,149,-153: 6661 patients were initiated and final success of FLMA was 6512.

The number do not add up to the figure of 6512. Based on the failures (89+60+14+46=209) it should be 6452. Please check.

In result section, from line 187 to line 193: The remaining 6661 patients were initiated with the FLMA, among whom 89 (1.3%) experienced primary failure after induction, and the primary success rate was 98.7%. 60 (0.9%) cases experienced secondary failure during surgical manipulation, among whom 14 (0.2%) occurred during surgical preparation when the position of the patient’s head and neck was changed, and 46 (0.7%) occurred during the intraoperative procedure. Final success of FLMA was achieved in 6512 cases (97.8%). Failures (89+14 +46=149). Success (6661-149=6512). Maybe in our original it does not describe the numbers clearly. We’ve checked the numbers and revised the description.

7. Table 1 only one p value (0.008) was given for all LMAs. Is it different for all sizes? 

Similarly for ASA grade, does one p value <0.001 applies to all grades?

Patients with and without FLMA failure were compared according to demographic and clinical characteristics by Pearson’s Chi-square test or Fisher’s exact test for categorical variables.(line 168 to line 169 in statistics section). In this study, we used Pearson’s Chi-square test or Fisher’s exact test to compare the successful rate among multiple groups (LMA size, or ASA grade). The statistic result can only show there is significant difference among the multiple groups. Thus, LMA size and ASA grade was considered a factor, and was entered into the logistic regression analysis. We didn't do Pairwise comparison to evaluate the specific differences between the two groups, which was unnecessary. Statistics method was from the relevance of Wang J, et al[4]. We’ve revised Table 1. 

8. Table 3 the total respiratory adverse events given as (n) 56 nerve. Adverse events under this subheading are they different to above total? If it is 56 then events mentioned do not add up. The subheading on tissue damage adverse events are immediate and delayed complications, suggest change of wording.

We’ve already change the complication to immediate respiratory adverse events and delayed complication. Both were airway complications in FESS in this study.

9 Discussion: under safety and efficacy: did all cases were compliant in maintaining the OLP and PIP difference of 5H20. Line 226 mentions inadequate relaxation, whether neuromuscular monitoring was carried out? 

Yes, all cases were compliant in oropharyngeal leak pressure (OLP) at least 5 cm H2O greater than peak inspiratory pressure (PIP), because .(method section, line 119 to line 120). Neuromuscular monitoring was not carried out for all cases and inadequate relaxation was our opinion. In our opinion, we consider that both surgical manipulation of the head and neck and insufficient muscle relaxation could lead to an abrupt increase in PIP, FLMA displacement and FLMA ventilation failure(line 288 to line 294). Because additional muscle relaxant could decrease the abruptly increased PIP in some of the medical chart.

10. Line 227- 233: Author’s mentions sometimes reducing tidal volume and increasing respiratory frequency were required. Is it author’s opinion on its management or speculation what occurred? 

It is author’s opinion on the anesthesia management. In order to reduce PIP, apart from additional muscle relaxant, we usually reduce tidal volume, by which we keep the difference between OLP and PIP. But it is our opinion, and we’ve cancelled it in the discussion section.

11. The line 246: previous study is on prevalence, risk factors, and management of asthma in China: a national cross-sectional study. I don’t understand the relevance here as this is FESS study assessing the safety and efficacy of the airway.

We want to compare the incidence (8.4%) of coexisting asthma observed in our study to a previous study of a Chinese adult population (4.2%). We want to show that the incidence of asthma are twice of the adult population.We’ve cancelled reference here.

12. Awake FLMA: what’s the relevance in your retrospective study? Has the authors conducted any such procedures?

It is not awake FLMA, is awake FLMA removal. But the sentence is confusing. We’ve changed to “Additionally, we suggest that the FLMA should be removed when the patient is fully conscious after FESS.” (line 346 to line 347)

13. Line 260 and 261 contradicts. If not documented in the medical chart, then is it author’s opinion?

Yes, it’s our opinion, so we use “our experience” to describe the reason of laryngospasm (line 335). Because the adverse event such as laryngospasm and bronchospasm and the relevant treatment is automatically record in the electronic anesthesia system, but the reason should be investigated later and manually added in the medical chart. Not all cases had the detailed reasons for laryngospasm in the medical chart. 

14. In results there were two patients had laryngospasm had NPPE. Describing their management in discussion may not be necessary?

Management has been cancelled.

15. Line 281: says no life threatening damage, but authors mentioned 5 such cases. It contradicts the previous statement. 

We’ve cancelled this statement here, and we thought the statement here was not appropriate. There were life-threatening damage cases in FESS under general in our study. But not all life-threatening complications are directly related to FLMA usage and we discussed them above. 

16. Limitations paragraph is mixed with limits and many improvement suggestions for the future. It’s confusing to the readers. 

We’ve re-organized our limitation from line 357 to 373, and the improvement suggestions for the future were put into the discussion section.

17. Line 301: ranks top: superlative may be avoided

It has been avoided

18. Conclusion: please refer to the comment in the abstract.

We’ve descrived the primary outcome and secondary outcome from line 381 to line 386.

Reference

1. Gao F, Li YR, Xu W, An YS, Wang HJ, Xian JF, et al. Upper airway morphological changes in obstructive sleep apnoea: effect of age on pharyngeal anatomy. J Laryngol Otol. 2020;134(4):354-61. PubMed PMID: 32284084.

2. Kim MS, Park JH, Lee KY, Choi SH, Jung HH, Kim JH, et al. Influence of head and neck position on the performance of supraglottic airway devices: A systematic r eview and meta-analysis. PLoS One. 2019;14(5):e0216673. PubMed PMID: 31071171.

3. Sanuki T, Uda R, Sugioka S, Daigo E, Son H, Akatsuka M, et al. The influence of head and neck position on ventilation with the i-gel airway in paralysed, anaestheti sed patients. European journal of anaesthesiology. 2011;28(8):597-9. PubMed PMID: 21505345.

4. Wang J, Shi X, Xu T, Wang G. Predictive risk factors of failed laryngeal mask airway insertion at first attempt. J Int Med Res. 2018;46(5):1973-81. PubMed PMID: 29569985.

---

## [Decision Letter · Decision Letter 1]

30 Nov 2020

PONE-D-20-20651R1

Safety, efficacy and airway complications of the flexible laryngeal mask airway in functional endoscopic sinus surgery: a retrospective study of 6661 patients

PLOS ONE

Dear Dr. Wang,

Thank you for submitting your manuscript to PLOS ONE. After careful consideration, we feel that it has merit but does not fully meet PLOS ONE’s publication criteria as it currently stands. Therefore, we invite you to submit a revised version of the manuscript that addresses the points raised during the review process.

We look forward to receiving your revised manuscript.

Kind regards,

Alkis James Psaltis, PhD, MBBS(HONS), FRACS

Academic Editor

PLOS ONE

Additional Editor Comments (if provided):

Thank you for your revised version.

Please address the questions addressed by reviewer 2 regarding Table 3 and the data entered.

Furthermore, the article as written really required a thorough grammatical and spelling review by a native English speaker to improve its readability to an acceptable standard

Reviewers' comments:

Reviewer's Responses to Questions

**Comments to the Author**

1. If the authors have adequately addressed your comments raised in a previous round of review and you feel that this manuscript is now acceptable for publication, you may indicate that here to bypass the “Comments to the Author” section, enter your conflict of interest statement in the “Confidential to Editor” section, and submit your "Accept" recommendation.

Reviewer #1: All comments have been addressed

Reviewer #2: (No Response)

2. Is the manuscript technically sound, and do the data support the conclusions?

Reviewer #1: Yes

Reviewer #2: Yes

3. Has the statistical analysis been performed appropriately and rigorously? 

Reviewer #1: Yes

Reviewer #2: Yes

4. Have the authors made all data underlying the findings in their manuscript fully available?

Reviewer #1: Yes

Reviewer #2: Yes

5. Is the manuscript presented in an intelligible fashion and written in standard English?

Reviewer #1: Yes

Reviewer #2: No

6. Review Comments to the Author

Reviewer #1: The authors have addressed the raised concerns of the first revision and edited the manuscript to a satisfactory degree

Reviewer #2: The query no 8 is not answered. Table 3 the total respiratory adverse events given as (n) 56 The events given do not add up to 56.

Many language or grammatical mistakes, which I cannot go thru each one. Example given below

If FLMA ventilation failed after three insertion attempts during the induction period,

ETT was immediately performed

May be reworded as: If FLMA ventilation failed after three insertion attempts during the induction period, patient was immediately intubated with ETT

Line 156 ETT was implemented

Suggest rewording: intubated with ETT

7. PLOS authors have the option to publish the peer review history of their article (what does this mean?). If published, this will include your full peer review and any attached files.

Reviewer #1: No

Reviewer #2: No

---

## [Author Response · Author response to Decision Letter 1]

23 Dec 2020

Dear Editor, 

I greatly appreciate you and the reviewers for all the comments, critiques and efforts to help optimize our manuscript. Both the editor and reviewers suggest that our manuscript requires a thorough grammatical and spelling review by a native English speaker to improve its readability. We asked American Journal Experts (AJE) to help us. Our manuscript has been edited by the highly qualified native English speaking editors at American Journal Experts (AJE), and the AJE certification has been uploaded a file. 

During the first communication, we think that we misunderstand the query No.8 of Reviewer #2, and we’ve made a better revision in this manuscript. We are happy to respond to the comments. 

Our point-by-point responses are listed below.

Reviewer #2:

1 Table 3 the total respiratory adverse events given as (n) 56 nerve. Adverse events under this subheading are they different to above total? If it is 56 then events mentioned do not add up. The subheading on tissue damage adverse events are immediate and delayed complications, suggest change of wording.

The query no 8 is not answered. Table 3 the total respiratory adverse events given as (n) 56 The events given do not add up to 56.

In the original manuscript, we thought the delayed complications such as sore throat and soft tissue injuries were respiratory adverse events. We think it is unappropriated and we’ve made a better Table 3 for respiratory adverse events, and the number adds up to 56.

2. Many language or grammatical mistakes, which I cannot go thru each one. Example given below. If FLMA ventilation failed after three insertion attempts during the induction period, ETT was immediately performed

Line 150 May be reworded as: If FLMA ventilation failed after three insertion attempts during the induction period, patient was immediately intubated with ETT

Line 156 ETT was implemented

Suggest rewording: intubated with ETT

Thank you for pointing out our language and grammatical mistakes. We should pay more attention on the fluency of our manuscript. We’ve asked the highly qualified native English speaking editors at American Journal Experts (AJE) for editing services, including grammar, phrasing and punctuations. We've revised many language and grammatical mistakes.

---

## [Editor Report · Decision Letter 2]

2 Jan 2021

Safety, efficacy and airway complications of the flexible laryngeal mask airway in functional endoscopic sinus surgery: a retrospective study of 6661 patients

PONE-D-20-20651R2

Dear Dr. Wang,

We’re pleased to inform you that your manuscript has been judged scientifically suitable for publication and will be formally accepted for publication once it meets all outstanding technical requirements.

Kind regards,

Alkis James Psaltis, PhD, MBBS(HONS), FRACS

Academic Editor

PLOS ONE

Additional Editor Comments (optional):

Thank you for your revisions. I think all of the issues have been adequately addressed
---

## [Editor Report · Acceptance letter]

14 Jan 2021

PONE-D-20-20651R2 

Safety, efficacy and airway complications of the flexible laryngeal mask airway in functional endoscopic sinus surgery: a retrospective study of 6661 patients 

Dear Dr. Wang:

I'm pleased to inform you that your manuscript has been deemed suitable for publication in PLOS ONE. Congratulations! Your manuscript is now with our production department. 

Kind regards, 

on behalf of

Dr. Alkis James Psaltis 

Academic Editor

PLOS ONE